# Proteomic Analysis of Circulating Extracellular Vesicles Identifies Potential Biomarkers for Lymph Node Metastasis in Oral Tongue Squamous Cell Carcinoma

**DOI:** 10.3390/cells10092179

**Published:** 2021-08-24

**Authors:** Xinyu Qu, Thomas C. N. Leung, Sai-Ming Ngai, Sau-Na Tsai, Abhimanyu Thakur, Wing-Kar Li, Youngjin Lee, Leanne Leung, Tung-Him Ng, Judy Yam, Linlin Lan, Eric H. L. Lau, Eddy W. Y. Wong, Jason Y. K. Chan, Katie Meehan

**Affiliations:** 1Department of Otorhinolaryngology, Head and Neck Surgery, The Chinese University of Hong Kong, Sha Tin, Hong Kong, China; xyqu@surgery.cuhk.edu.hk (X.Q.); leanneleung@ent.cuhk.edu.hk (L.L.); linlinlan@ent.cuhk.edu.hk (L.L.); ericlau.hl@gmail.com (E.H.L.L.); eddywywong@yahoo.com.hk (E.W.Y.W.); jasonchan@ent.cuhk.edu.hk (J.Y.K.C.); 2State Key Laboratory of Agrobiotechnology, School of Life Sciences, The Chinese University of Hong Kong, Sha Tin, Hong Kong, China; chunningtleung@cuhk.edu.hk (T.C.N.L.); smngai@cuhk.edu.hk (S.-M.N.); helentsai@cuhk.edu.hk (S.-N.T.); 3Department of Neuroscience, Jockey Club College of Veterinary Medicine and Life Sciences, City University of Hong Kong, Kowloon Tong, Hong Kong, China; abithakur1211@gmail.com (A.T.); younglee@cityu.edu.hk (Y.L.); 4Department of Biomedical Sciences, Jockey Club College of Veterinary Medicine and Life Sciences, City University of Hong Kong, Kowloon Tong, Hong Kong, China; wingkarli2-c@my.cityu.edu.hk; 5Ben May Department for Cancer Research, Pritzker School of Molecular Engineering, The University of Chicago, 929 East 57th Street, Chicago, IL 60637, USA; 6Department of Pathology, Li Ka Shing Faculty of Medicine, The University of Hong Kong, Pok Fu Lam, Hong Kong, China; tonyng93@hku.hk (T.-H.N.); judyyam@pathology.hku.hk (J.Y.)

**Keywords:** extracellular vesicles, oral tongue squamous cell carcinoma, lymph node metastases, proteomics, biomarkers

## Abstract

Lymph node metastasis is the most reliable indicator of a poor prognosis for patients with oral tongue cancers. Currently, there are no biomarkers to predict whether a cancer will spread in the future if it has not already spread at the time of diagnosis. The aim of this study was to quantitatively profile the proteomes of extracellular vesicles (EVs) isolated from blood samples taken from patients with oral tongue squamous cell carcinoma with and without lymph node involvement and non-cancer controls. EVs were enriched using size exclusion chromatography (SEC) from pooled plasma samples of patients with non-nodal and nodal oral tongue squamous cell carcinoma (OTSCC) and non-cancer controls. Protein cargo was quantitatively profiled using isobaric labelling (iTRAQ) and two-dimensional high-performance liquid chromatography followed by tandem mass spectrometry. We identified 208 EV associated proteins and, after filtering, generated a short list of 136 proteins. Over 85% of the EV-associated proteins were associated with the GO cellular compartment term “extracellular exosome”. Comparisons between non-cancer controls and oral tongue squamous cell carcinoma with and without lymph node involvement revealed 43 unique candidate EV-associated proteins with deregulated expression patterns. The shortlisted EV associated proteins described here may be useful discriminatory biomarkers for differentiating OTSCC with and without nodal disease or non-cancer controls.

## 1. Introduction

It is estimated that ~380,000 people were diagnosed and ~180,000 died from oral cancer worldwide in 2020 [1]. For oral tongue squamous cell carcinoma (OTSCC), early-staged OTSCC has a better survival, with stage I disease having an 80% five-year overall survival rate as opposed to 15% for stage IV disease [2]. Unfortunately, these types of cancers are routinely discovered at an advanced stage as they can develop without producing pain or symptoms and are associated with a poor prognosis. Surgical therapy is morbid, particularly for more advanced stages which require extensive resections and free tissue transfer for reconstruction, with a consequent negative impact on swallowing, speech and physical appearance [3]. This, combined with the need for additional radiotherapy ± chemotherapy, results in significant morbidity [4].

Lymph node metastasis not only increases the possibility of distant metastasis, but also has a negative impact on overall survival [5]. Currently, advanced disease characterized by metastatic nodal involvement is the only reliable clinical indicator of poor prognosis for patients with OTSCC. However, this is difficult to assess pre-operatively with imaging as metastases can occur microscopically and can spread in an unpredictable anatomical manner. Further, there are currently no biomarkers to predict whether a cancer will spread in the future if it has not already spread at the time of diagnosis. The need for better biomarkers and an improved understanding of the mechanisms governing oral cancer progression is paramount to realize more robust therapeutic strategies.

Recent studies have shown that extracellular vesicles (EVs) secreted by tumor cells are key messengers for inducing tumor vascular permeability, inflammation, and recruitment of bone marrow progenitor cells in remote organs for cellular communication [6,7,8]. For example, EVs containing EGFR regulate the liver microenvironment and promote liver metastasis [9], EV miR-25-3p is involved in the formation of a pre-metastatic niche before metastasis [10], and EV integrin expression facilitates tumor homing to specific distant metastatic sites [11]. Supporting this, proteomic analyses of plasma EVs from patients with papillary thyroid cancer with and without lymph node metastases revealed overexpression of proteins known to be involved in metastasis, such as proto-oncogene Tyrosine-protein Kinase Src, Integrin Beta 2, and Calpain small subunit 1 [12]. A similar study on crude plasma EVs from patients with oral cancer with and without lymph node metastases revealed deregulated expression of several typical proteins involved in the establishment of a pre-metastatic niche [13]. Although some potentially significant relationships with clinic-pathological features were cited, previous studies have suffered from some important limitations and follow-up studies are lacking. Based on the potential role of EVs as mediators of tumorigenesis, the aim of this work was to perform a quantitative proteomics analyses of purified, host-derived plasma EVs from a homogeneous cohort of OTSCC patients with and without nodal disease.

## 2. Materials and Methods

### 2.1. Participants, Ethics, and Consent

Ten non-cancer controls and fourteen patients with OTSCC were recruited at Prince of Wales Hospital, Hong Kong following ethical approval from the Joint Chinese University of Hong Kong—New Territories East Cluster Clinical Research Ethics Committee (reference number 2015.396). Study participants provided written informed consent prior to study enrolment. Specimens were obtained prior to commencement of treatment. OTSCC patients were included in this study if they were >18 years old; had a histologically confirmed diagnosis of a primary squamous cell carcinoma in the oral tongue; resectable disease with or without nodal involvement; and no prior systemic therapy. Exclusion criteria were an Eastern Cooperative Oncology Group performance status of ≥2; prior history of radiation therapy to the head and neck region; primary tumor site other than oral tongue; pregnant or breastfeeding; rapidly progressing disease; or known history of human immunodeficiency virus or active Hepatitis B or C. Non-cancer controls were included if they were over 18 years of age and had no history of cancer.

### 2.2. Extracellular Vesicle Isolation

Whole blood from OTSCC patients was collected in EDTA Vacutainer^®^ blood collection tubes (Becton Dickinson, Franklin Lakes, NJ, USA) and processed within 2 h of collection. First, blood was centrifuged at 1600× *g* for 20 min. The plasma fraction was further clarified by two rounds of centrifugation at 16,000× *g* and 10,000× *g* for 10 min each. We prepared three plasma pools prior to EV isolation by combining 200 µL of plasma from each group. This resulted in plasma pools with a total volume of 2 mL for non-cancer controls, 1.6 mL for non-nodal OTSCC, and 1.2 mL for nodal OTSCC. The same volume of pooled plasma (900 μL) from each group was then diluted with 1.1 mL filtered PBS prior to isolation using a qEV2/35 nm size exclusion column (Izon Science, Christchurch, New Zealand). Fractions 8 to 10 were pooled based on the presence of the known EV marker, TSG101 as determined by Western blot, EV, and protein concentration. Pooled fractions were concentrated to 100–200 µL using Amicon^®^ Ultra-15 (Merck, Darmstadt, Germany) filtration units. Concentrated EV samples were stored at −20 °C.

### 2.3. Transmission Electron Microscopy (TEM)

Fresh EV samples were fixed in 2.5% glutaraldehyde overnight and transferred onto 200 mesh Formvar-carbon coated copper grids (Electron Microscopy Sciences, Hatfield, PA, USA). Following staining in 2% phosphor-tungstic acid staining, grids were dried overnight and visualized using the Hitachi HT7700 electron microscope (Hitachi, Tokyo, Japan) at an operating voltage of 110 kV.

### 2.4. Nanoparticle Tracking Analysis (NTA)

EV samples (100 μL) were diluted 1/200 in filtered PBS and injected into the analysis chamber of the NanoSight NS300 Instrument equipped with a 488 nm laser and an sCMOS camera (Malvern Panalytical, Malvern, UK). Sample analysis was performed at a camera level of 10 and gain of 250, with a detection threshold of 10 pixels. Settings for blur, minimum track length, and minimum expected size were set to ‘auto’. Videos were recorded for 60 s at 30 frames/s in triplicate at 25 °C. All post-acquisition settings remained constant between samples. NTA software v3.0 was used to process and analyze the data.

### 2.5. Western Blot

EVs were lysed with RIPA buffer (Merck, Darmstadt, Germany) supplemented with protease inhibitors (Roche, Mannheim, Germany) for 1 h on ice and centrifuged at 12,000× *g* for 20 min. After Pierce™ BCA protein assay (Thermo Fisher Scientific, Waltham, MA, USA), 10 µg of proteins were diluted in loading buffer (30% glycerol, 10% SDS, 0.012% bromophenol blue) and denatured at 95 °C for 10 min. Proteins were electrophoresed and transferred onto a Nitrocellulose Membrane (Bio-Rad, Hercules, CA, USA) using the Mini-PROTEAN Tetra Vertical Electrophoresis Cell (Bio-Rad) at a constant voltage of 110 V for 2 h. The membrane was blocked for 60 min in 5% non-fat milk and primary antibodies were diluted 1:1000 as follows: mouse anti-human CD9 (clone MM2/57, Invitrogen, Waltham, MA, USA), mouse anti-human Calnexin (clone AF18, Invitrogen), and mouse anti-human TSG101 (clone 51, BD Transduction Laboratories, Franklin Lakes, NJ, USA). After overnight incubation with primary antibody, secondary anti-mouse IgG, HRP-linked antibody (Cell Signaling Technology, Danvers, MA, USA) was diluted 1:2000 and added for 60 min. Signals were developed using Clarity™ Western ECL Blotting Substrates (Bio-Rad) and were imaged using the ChemiDoc™ Touch Imaging System (Bio-Rad). Images were processed using Image Lab™ software v6.0 (Bio-Rad).

### 2.6. Extracellular Vesicle Protein Digestion and Labelling

Following protein determination by the Pierce™ BCA assay (Thermo Fisher Scientific), 30 μg of EV protein preparations from the non-cancer control, non-nodal OTSCC, and nodal OTSCC plasma pools were prepared as follows. Air dried protein pellets were prepared for MS by dissolving in 6 M urea and 2 M thiourea containing protease and phosphatase inhibitor cocktail (Thermo Fisher Scientific). The cysteine residues were reduced by adding 5 mM *tris*-(2-carboxyethyl) phosphine (Sigma, St. Louis, MO, USA) for 1 h at 60 °C, and alkylated by adding 10 mM methyl methanethiosulfonate (Sigma) for 10 min in the dark at room temperature. The concentration of urea was reduced below 0.75 M with 50 mM triethylammonium bicarbonate (TEAB), pH 7.8 (Sigma). Proteins were digested with sequencing-grade trypsin (Promega, Madison, WI, USA) at a ratio of 1:10 overnight at 37 °C. Peptide concentrations were determined using a Pierce™ quantitative colorimetric peptide assay (Thermo Fisher Scientific). Peptides were dried using a CentriVap benchtop vacuum concentrator (Labconco, Kansas City, MO, USA), and reconstituted in 50 mM TEAB, pH 7.8. Reconstituted peptides were then processed according to the manufacturer’s protocol for isobaric tags for relative and absolute quantitation (iTRAQ) 8 plex (Sigma, St. Louis, MO, USA) for comparative and quantitative analyses (iTRAQ-8plex labels; non-cancer controls = 118, non-nodal OTSCC = 114, nodal-OTSCC = 117). Samples were mixed and desalted using C18 spin columns (Thermo Fisher Scientific) prior to nanoLC-MS/MS.

### 2.7. Liquid Chromatography Tandem Mass Spectrometry Analysis

iTRAQ-labelled peptides were subjected to Dionex UltiMate 3000 RSLC nano system interfaced to a Orbitrap Fusion Lumos Tribrid mass spectrometer (Thermo Fisher Scientific) equipped with a Nanospray Flex ion source. Peptides were loaded onto a trapping column (Acclaim™ PepMap™, 0.3 mm × 5 mm, nanoViper fitting C18, 5 µm, 100 Å, Thermo Fisher Scientific) for pre-concentration, and then resolved in the analytical column (Acclaim™ PepMap™, 75 µm × 25 cm, nanoViper fitting C18, 2 µm, 100 Å, Thermo Fisher Scientific) for peptide separation. The separation of peptides was achieved at a constant flow rate of 300 nL/min, using a linear gradient from 2% to 5% of solvent B (acetonitrile (ACN), 0.1% TFA) over 3 min and then from 5% to 55% of ACN over 180 min. Ions were generated by positive electrospray ionization via liquid junction into the mass spectrometer. Mass spectra were acquired over *m*/*z* 375–1500 at 120,000 resolution. Ions were selected by data-dependent acquisition with Top Speed method of 3 s cycle time for tandem mass spectrometry by HCD fragmentation, fixed collision energy of 38%, and a resolution of 15,000. A 1.2 *m*/*z* isolation window and a fixed first mass of 100 *m*/*z* were used for MS/MS scans. Automatic gain control targets were 4E5 ions for MS scans and 5E4 for MS/MS scans. Dynamic exclusion was set at 60 s. Rejection of precursor ions with charge state +1, greater than +8, and unassigned charge features were employed to minimize redundant MS/MS collection and maximize peptide identifications.

### 2.8. Data Analysis

Mass spectrometry data were analyzed using Proteome Discoverer version 2.4 (Thermo Fisher Scientific) with SEQUEST as the search engine. The protein false discovery rate was estimated with Percolator using an experimental *q*-value (exp. *q*-value) threshold of 0.05. Data were searched against UniProt *Homo sapiens* database. The searching parameters were as follows: iTRAQ-8plex of lysine, tyrosine, or peptide N-terminus (+144.102 Da), methyl-thiol of cysteine (+45.988 Da), and oxidation of methionine (+15.9949 Da) was set as dynamic modification; precursor-ion mass tolerance, 10 ppm; fragments-ion mass tolerance, 0.02 Da. Trypsin was set as the digestion enzyme with two missed cleavages permitted. Protein localization and functions were interrogated using FunRich (version 3.1.3) and String (version 11.0) [14,15]. Survival analysis was performed using UALCAN or Kaplan-Meier plotter and the head and neck cancer TCGA dataset [16,17,18].

### 2.9. Statistical Analysis

Comparison of EV levels between groups was made using the Kruskal–Wallis one-way analysis of variance test and a *p* value of <0.05 was considered statistically significant. Statistical analysis was conducted using SPSS version 26.0 (IBM, Armonk, NY, USA).

## 3. Results

### 3.1. Patient and Clinical Characteristics

A total of fourteen patients with OTSCC and ten non-cancer controls were included in this study (*n* = 24). Patient and tumor characteristics are outlined in Table 1. The median age was 61 (range 52–74) and 64% of the patients were male (9/14). In the non-nodal OTSCC patient cohort, all patients T stage 1 or 2 and most were non-drinkers (62%) and non-smokers (62%). In the nodal OTSCC patient cohort, half were T stage 4a and again, most were non-drinkers (67%) but only half were non-smokers (50%). The median age of non-cancer controls was 58 (range 23–83) and 50% were male. All non-cancer controls were non-drinkers and 80% were non-smokers.

### 3.2. Validation of Isolated EVs

Plasma EVs of patients with non-nodal and nodal OTSCC and non-cancer controls were enriched using size exclusion chromatography (SEC) from pooled plasma samples (Section 2) (Figure 1A). NTA and protein quantitation showed that fractions 8–10 contained the highest number of particles with the lowest protein concentration (Figure 1B). Western blot analysis confirmed that these fractions contained TSG101 positive EVs (Figure 1C). NTA analysis of EVs from the pooled plasma samples showed the mean size was approximately 130 nm (range 60–540 nm) and the concentration was approximately 3.09 × 10^11^ particles/mL of plasma (range 2.26 × 10^10^–6.30 × 10^11^ particles/mL plasma) (Figure 1D). Consistent with previous reports, the mean particle size and concentration of EVs from the non-cancer control group, OTSCC without nodal disease, and OTSCC with nodal disease groups were 151.9, 147.0, 183.1, and 2.26 × 10^10^ particles/mL, 2.74 × 10^11^ particles/mL, and 6.30 × 10^11^ particles/mL, respectively [13,19]. Consistent with earlier reports, we report that higher levels of plasma EVs were associated with nodal disease burden compared with OTSCC without nodal disease and non-cancer controls (*p*-value = 0.02) (Figure 1G). The mean protein concentration of EVs from the pooled plasma samples was higher in the OTSCC groups (0.39 and 0.43 µg/µL for non-nodal and nodal disease respectively) compared with non-cancer controls 0.21 µg/µL) however, the differences were not statistically significant. The NTA data were corroborated by TEM data showing spherical particles with a bilayer membrane in the size range of 100–180 nm (Figure 1E). Western blot analysis of EVs from pooled plasma samples detected CD9, a known EV protein biomarker according to internationally accepted guidelines (Figure 1F) [20]. Though CD9 was detected by Western blot, along with some other classic EV markers, it was not detected by mass spectrometry. This may be due to its hydrophobic nature and large molecular size—features that may prohibit mass spectrometry analyses depending on the sensitivity of the instrument used and the sample input [21]. We highlight that Calnexin, an endoplasmic reticular protein, was not detected in the EV samples by Western blot or mass spectrometry. This provides additional evidence that the isolated EVs were purified and not contaminated by redundant intracellular components. Building on mounting literature, we were unable to routinely identify select tetraspanins (CD63/81) in plasma EVs [22]. We echo previous reports that tetraspanins are not universal markers for EVs isolated from plasma. In aggregate, these results demonstrate that EVs were successfully isolated from the pooled plasma samples with high purity and well characterized by various methods.

### 3.3. Quantitative Extracellular Vesicle Proteomic Profiling

Isolated EVs (Figure 1) were subjected to tryptic digestion, isobaric labelling (iTRAQ), and quantitative proteomics which identified 208 unique proteins. Of these, 2 proteins were excluded based on a q-value which was greater than 0.05 and considered to be an insignificant change in expression, 69 were excluded due to missing labels and incomplete quantitative data, and 1 trypsin protein was excluded (Appendix A). For the remaining 136 proteins, raw abundance values of each protein were converted into a ratio by dividing reporter-ion abundance values between comparison groups. Proteins with an empirical fold change of greater than 1.5 were considered de-regulated. Both high-abundant plasma proteins (such as albumin, fibrinogen, haptoglobin, alpha-2-Macroglobulin, and various immunoglobulins) and classical EV proteins (such as glypican-1, CD44, and von Willebrand factor) were identified. Albumin and immunoglobulins are two of the most abundant plasma proteins, and their detection not only implies plasma protein contamination, but may also have reduced the sensitivity of the mass spectrometry (MS) by narrowing its dynamic range of analysis. On the other hand, glypican-1, CD44, and von Willebrand factor were used as indicators of plasma EV enrichment. CD44 is a cell surface glycoprotein while glypican-1 is a membrane-anchored proteoglycan and both have been reported to be enriched in EVs [23,24]. von Willebrand factor is one of the most common protein markers found in platelet-derived EVs [25,26], which accounts for up to 90% of EVs found in plasma. Due to the classical, high abundant plasma protein nature of 59 proteins, these have been analyzed as a separate sub-group in order to maximize the capacity of the analysis consistent with a recent study [27]. The remaining 77 EV associated proteins of interest were compared between the non-cancer control group, the OTSCC group without nodal disease and the OTSCC group with nodal disease.

### 3.4. Analysis of EV-Associated Proteins of Interest

EV-associated proteins (*n* = 77) were compared with the Vesiclepedia database to reveal that 96% had been identified as EV associated previously (Figure 2A). Funrich analysis verified this by showing that most proteins associated with the terms exosomes or extracellular when categorized according to the cellular compartment with which they associate (Figure 2B). Most proteins (>85%) were associated with exosomes and the extracellular space but some were also categorized as associated with the lysosome or cytoskeleton. As lysosome and cytoskeletal related proteins are unlikely to be present in EVs, it is possible that we have co-isolated extra-vesicular proteins. Nevertheless, the putative EV protein cargo were also categorized according to their role within biological pathways, a strategy that identified the complement cascade and integrin signaling as the most overrepresented within these vesicles (Figure 2C).

Analysis of proteins showing altered abundance between the non-cancer controls, OTSCC without nodal disease, and OTSCC with nodal disease revealed 43 (56%) candidate proteins (Table 2) based on deregulated expression between groups. Based on GO ontology, these proteins are overrepresented in complement, extracellular matrix, and integrin signaling. The expression patterns suggest that EV-associated platelet factor 4 variant, tubulin beta-4A chain, histone H2B type 2-E and collagen alpha-1(I) may be potential markers of OTSCC and may also be informative of nodal status. In comparison, EV-associated Syndecan-1 and 40S ribosomal protein SA protein expression may be markers of OTSCC but are unlikely to be informative of nodal status. EV-associated proteins that may be useful for identifying OTSCC with nodal disease include integrin beta-4, cathelicidin antimicrobial peptide, multimerin-1, and others. In contrast, we identified 26 EV-associated proteins that may be useful for identifying OTSCC without nodal disease including serum amyloid A-4 protein, serum paraoxonase/arylesterase 1 and tumor-associated calcium signal transducer 2. To investigate whether there is any similarity between tumor expression and EV cargo, an in silico analyses was performed. According to the Human Protein Atlas, about 60% of these EV-associated proteins (25 out of 43) are expressed in head and neck tumors based on immunohistochemistry. This is consistent with previous reports suggesting that the levels of proteins packaged into plasma derived EVs may not necessarily correlate with their relative intracellular concentration or tissue expression [28]. The mediocre correlation between tumor and EV protein expression may be due to the transient biology of an evolving tumor or it may reflect the dynamic nature EV packaging. Regardless, additional in silico analysis revealed that high tumoral expression of myosin-9, agrin, integrin alpha-3, actin, and serum paraoxonase/arylesterase 1 alone and combined were significantly associated with shorter overall survival (Figure 3). Collectively, these findings suggest that EV-packaging of these proteins may be worthy of further investigation.

## 4. Discussion

Neck lymph node positivity in OTSCC correlates with poor prognosis but is difficult to predict. There is an unmet need for sensitive and specific prognostic biomarkers. This study has identified several potential novel, non-invasive candidate biomarkers in plasma EVs that are associated with lymph node disease. Previous oral cancer biomarker studies investigating plasma proteins have failed to deliver clinically translational results [30]. This is likely due to the large dynamic range of protein concentrations in plasma and the fact that only 12 highly abundant proteins constitute >95% of total plasma proteins [31]. With most studies, these high abundant species generally prohibit detection of the lower abundant species in crude plasma. The discovery of EVs as a functionally relevant biomarker resource has revolutionized our capacity to fractionate plasma and dig deeper into the proteome. Building on the notion that the tumor mass is responsible for high levels of circulating EVs detected in cancer patients, we showed that higher levels of plasma EVs were associated with nodal disease burden. While this trend has been shown in previous reports, it remains to be established whether host-derived, plasma EV levels alone offer a robust biomarker of disease.

Beyond EV levels, we identified a set of 43 candidate EV proteins that were deregulated in plasma EVs from patients with OTSCC. Of these 4 proteins (platelet factor 4 variant, tubulin beta-4A chain, histone H2B type 2-E, and collagen alpha-1(I)) may be potential markers of OTSCC and may also be informative of nodal status. EV packaged platelet factor 4 variant has been identified in plasma from patients with oral cancer previously underscoring the potential importance of this as a biomarker [13]. In addition, deregulated tumor tissue gene expression of Platelet factor 4 variant 1 has been reported previously [17]. This chemokine-like protein displays strong anti-angiogenic and tumor suppressive capacities and appears to correlate with the differentiation degree of oral cancers and the number of positive nodes. Pre-clinical studies have shown that platelet factor 4 variant 1 can prevent nodal metastases by inhibiting tumor lymphangiogenesis, cell migration, tumor growth, and angiogenesis [32,33,34]. In keeping with this concept, we and others report reduced expression of platelet factor 4 variant 1 in oral cancers with nodal disease, and speculate that its downregulation is a deliberate mechanism adopted by OTSCC to facilitate nodal metastases. This underscores the potential of this protein as a biomarker for OTSCC.

Out data suggest that syndecan-1 and 40S ribosomal protein SA may be potential biomarkers of OTSCC but neither are unlikely to be informative of nodal status. Consistent with up-regulated levels of syndecan-1 in OTSCC reported here, previous studies have shown that this protein promotes oral tongue cancer cell migration [35] and is indicative of the presence of distant metastases [36]. However, these findings are contrary to earlier reports suggesting that reduced tissue expression of syndecan-1 in head and neck cancer correlates with the progression of carcinogenesis, grade, and tumor size [37,38,39]. This maybe an example where tissue expression may not correlate with circulating levels of a protein. Whether EV packaging of syndecan-1 is a mechanism used by OTSCC to promote migration and other known roles of the syndecan-1 cell surface proteoglycan family (proliferation, inflammation, angiogenesis, and tumorigenesis) warrants further investigation [40,41]. Regardless of its function, high levels of EV packaged syndecan-1 may be a useful candidate biomarker for OTSCC.

Of the 11 EV-associated proteins that may be useful markers of nodal OTSCC, transferrin receptor protein 1 and integrin beta-1 have been identified by previous EV head and neck cancer studies [13,19]. In general, we found that integrin signaling was one of the most highly represented protein family in this analysis, albeit down-regulated in OTSCC. While EV expression of integrin’s correlated with metastatic tropism in other cancer types [11], our data do not support this phenomenon in OTSCC. This may be due to the unique tumor biology associated with OTSCC. Up to 50% of patients with locally-advanced OTSCC may develop recurrence but fewer than 10% of cases experience distant relapse [42]. That is, the majority of OTSCC recurrences occur in regional lymph nodes. As such, we propose that EV-integrin expression may be down-regulated in the cohort of OTSCC included in this analysis because distant metastases were not evident. EV-integrin directed organotrophism may not play as critical a role in this cohort or cancer type but further studies including cases with distant metastases may suggest otherwise.

We also identified 26 EV-associated proteins that may be useful for identifying OTSCC without nodal disease. These include serum amyloid A-4, complement C1q subcomponent subunit C, and von Willebrand factor, all of which have been identified by previous EV head and neck cancer studies [13,19]. Complement C1q is the first recognition subcomponent in the classical pathway of the complement system and was the most highly deregulated protein detected in this study. For subcomponent B of C1q, we observed an almost 6-fold increase in EVs from patients with non-nodal OTSCC compared with non-cancer controls and then an 8-fold decrease in expression in nodal OTSCC compared with non-nodal disease. A similar but less dramatic trend was observed for subcomponent C. C1q is a promiscuous protein and can perform various immune and non-immune functions in complement dependent or independent manners. Various studies have shown that C1q may play dual survival-promoting and apoptosis-inducing roles in cancer, but this is most likely borne out by the type of cancer and the tumor-immuno-microenvironment [43,44,45,46]. Therefore, it is not surprising that we observed major differences in EV expression of C1q between nodal and non-nodal OTSCC.

A previous study identified EV proteins from plasma of patients with head and neck cancers that were associated with the response to chemo-radiation, but was constrained by certain important limitations [19]. Firstly, this study was based on data from pooled plasma of a small cohort (*n* = 12) of patients with head and neck cancers originating from a range of sub-sites (oral cavity, oropharynx, and larynx) that are known to be associated with unique etiologies. Secondly, targeted proteomics was performed and a small percentage of the proteome (~4%, 656 out of ~18,000 proteins) was assessed [47]. Finally, specialized, sub-fractions of EVs (cholerae toxin chain B^+^ and annexin V^+^ EVs) were analyzed as opposed to host-derived, all-inclusive EVs or cancer-specific EVs which are currently challenging to isolate. Despite these constraints and differences in study design compared with our work, we highlight that both this study and ours identified upregulated expression of von Willebrand factor and integrin beta-1 in EVs from patients with head and neck cancer [19]. The former study identified elevated levels of these proteins in non-responders to chemo-radiation whereas we identified elevated levels of these proteins in OTSCC patients without nodal disease.

Another study identified four proteins (platelet factor 4 variant 1, C-X-C motif chemokine, cDNA FLJ93141, and apolipoprotein A-1) that may be related to the metastasis of oral squamous cell carcinoma, and it was suggested that they may be helpful for the diagnosis of lymph node metastases [13]. This study is exciting and generated some potentially, clinically relevant results but also suffered from some limitations. Although reasonably sized test (*n* = 30) and validation (*n* = 60) cohorts were studied, beyond classifying the cases as oral squamous cell carcinomas, the precise origin of the primary tumor sub-site was not disclosed and the heterogeneity of the cohort is unclear. Again, this is important considering the vast difference in etiologies associated with head and neck cancers occurring in different anatomical sites. Further, EVs were isolated using a crude precipitation method that tends to co-isolate non-EV particles, proteins, and other circulating components. Regardless of these limitations, both this study and ours identified von Willebrand factor, complement C1q subcomponent subunit C, transferrin, platelet factor 4 variant 1, and serum amyloid A-2 protein. We both reported similar expression trends for plasma EV levels of transferrin (reduced expression in nodal OTSCC relative to non-cancer controls), serum amyloid A-2 protein (reduced expression in nodal OTSCC relative to non-nodal disease), and partially similar trends for platelet factor 4 variant 1 (increased expression in non-nodal OTSCC and reduced expression in nodal OTSCC relative to non-cancer controls). However, our results conflicted with the trends reported for von Willebrand factor and complement C1q subcomponent subunit C.

Along with an increasing collection of publications, we report the detection of EV associated histone proteins [29,48,49]. While it is possible that these proteins may be surface-associated or extra-vesicular, their regulated expression in EVs from patients with OTSCC is intriguing. Here, we show that EV associated histone proteins were increased in OTSCC without nodal disease and decreased nodal disease relative to non-cancer controls. Beyond the biomarker potential, the biological ramifications of this are worthy of consideration. Histone variants and their chaperones are altered in solid tumors and play a fundamental role in compacting DNA into nucleosomes to aid DNA packaging and cellular proliferation [50]. Based on the premise that EV packaged proteins directly reflects their cell of origin by mimicking expression or action, it is curious that EV associated histones are increased in non-nodal OTSCC yet decreased in nodal disease. Increased EV associated histone expression in non-nodal OTSCC may reflect an early onco-promoting event that is associated with disease establishment rather than metastases.

An earlier study suggested that levels of glypican 1 expressing EVs could discriminate between early- and late-stage pancreatic cancer, benign pancreatic disease and non-cancer controls [23]. A more recent study failed to verify this and is supported by our own data [51]. Here, we show that EV associated glypican 1 expression was similar in plasma from non-cancer controls and OTSCC regardless of nodal status. A logical explanation for our discrepant results may be the unique tumor biology associated with OTSCC in contrast to pancreatic cancer. This finding highlights the need for careful interpretation of big “omics” datasets originating from different cancer types.

Overall, we have shown that there are a large number of deregulated EV proteins in plasma from patients with and without nodal OTSCC compared with non-cancer controls. We highlight that this is in stark comparison with our tissue-based gene expression study which showed negligible changes in primary tissues of OTSCC with and without nodal disease and non-cancer controls [52]. Although our earlier study utilized tissues and targeted immuno-cancer genes and the current study utilized plasma and was untargeted, we expected some overlap. While we have identified set of bona fide EV components, this study suffered from some limitations. While homogeneous, the cohort was small and pools of plasma were used. Future studies incorporating statistically powered, larger cohorts are needed. Further, the coverage of the EV proteome could be improved upon further depletion of co-purified high abundant serum proteins and increased chromatographic fractionation. Finally, although current methods are under development, future studies should strive to profile single populations of EVs or cancer-specific EVs. Nevertheless, we are excited by our hypothesis-generating results and propose that the shortlisted EV associated proteins may be useful discriminatory biomarkers for differentiating OTSCC with and without nodal disease or non-cancer controls.

## Figures and Tables

**Figure 1 cells-10-02179-f001:**
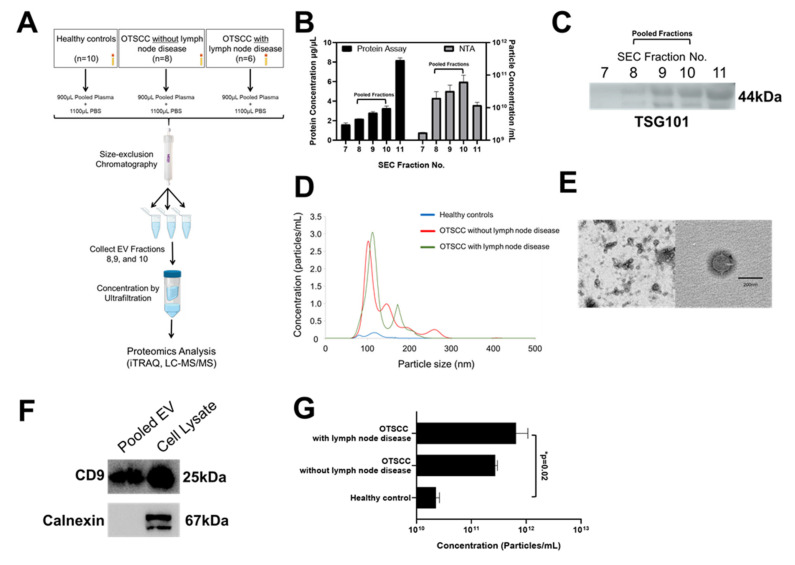
Validation of extracellular vesicle isolation using NTA, TEM, and Western blotting. (**A**) Overview of work flow. (**B**) NTA and protein quantitation indicated the mean particle concentration and crude protein concentration of EVs in SEC fractions 7–11. (**C**) Western blot results demonstrated the abundance of the EV positive marker TSG101 in SEC fractions 7–11. (**D**) NTA indicated the mean size distribution and particle concentration of EVs in each group. (**E**) Representative TEM image showed the characteristic rounded structure of EVs that are of the expected size. (**F**) Western blot results demonstrated the abundance of the EV positive marker CD9 SEC pooled fractions as well as whole cell lysate of an oral tongue cancer cell line (SCC25). Calnexin was negative in the EV preparations but positive in the cell lysate as expected. (**G**) Particle concentration was significantly greater in plasma from patients with node positive OTSCC relative to healthy controls. There was a reduction of EVs between OTSCC with and without nodal disease, but this trend was not significant.

**Figure 2 cells-10-02179-f002:**
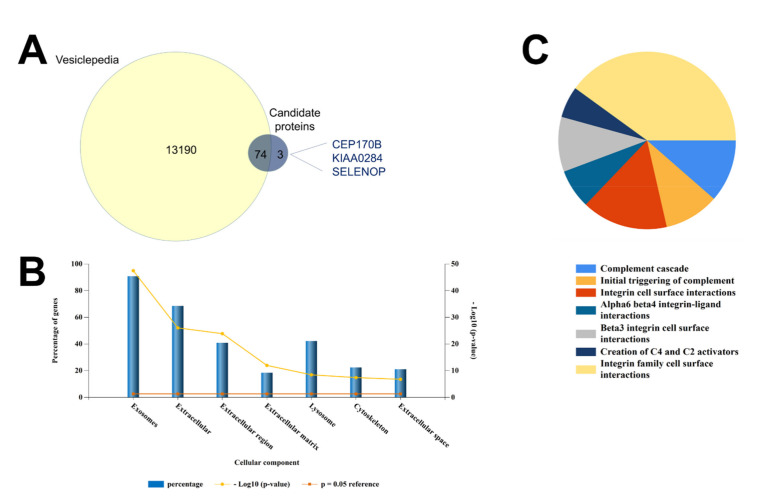
Cellular and biological pathways that are over-represented among extracellular vesicle associated proteins. (**A**) EV data were compared with Vesiclepedia to verify that most proteins identified have been previously reported as EV-associated. (**B**) EVs/exosomes were the most enriched cellular components. The *x* axis on the left represents the percentage of proteins (blue bars) identified in our data set that are expressed or involved in each category. The *x* axis on the right represents the significance of this as log 10 (*p*-value) (yellow line). The red line represents the significance at *p* = 0.05 as a reference. (**C**) The complement cascade and integrin signaling were the most overrepresented biological pathways.

**Figure 3 cells-10-02179-f003:**
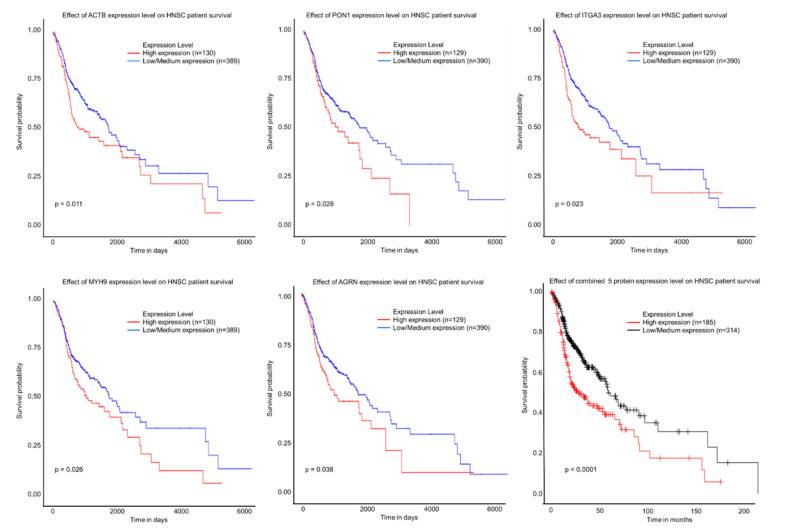
Kaplan-Meier analyses showing the relationship between survival and head and neck tumoral expression of candidate EV associated proteins (actin, serum paraoxonase/arylesterase 1, integrin alpha-3, myosin-9, and agrin). Survival curves of individual proteins were generated using UALCAN and the effect of combined expression of all 5 proteins was generated using Kaplan-Meier plotter [18]. Similar to many other proteomic studies, we identified a number of classical, high abundant, plasma proteins [13,19,29]. Quantitative analyses of this sub-group of EV associated proteins revealed that a lower percentage (37%, 22 out of 59) changed in abundance between groups when compared with candidate EV proteins (Table 3). Intriguingly, apolipoproteins were increased in non-nodal OTSCC relative to non-cancer controls but decreased in nodal OTSCC relative non-nodal disease. Similar trends were observed for some immunoglobulins, keratin and serrotransferrin.

**Table 1 cells-10-02179-t001:** Patient and tumor characteristics.

Characteristics	Total	Non-Nodal OTSCC	Nodal OTSCC
	(*n* = 14)	(*n* = 8)	(*n* = 6)
Age			
Median (range) years	61 (52–74)	62 (58–74)	60 (53–65)
Gender			
Female	5 (36%)	4 (50%)	1 (17%)
Male	9 (64%)	4 (50%)	5 (83%)
T Stage			
1	7 (50%)	7 (87.5%)	0 (0%)
2	3 (21.5%)	1 (12.5%)	2 (33%)
3	0 (0%)	0 (0%)	0 (0%)
4	1 (7%)	0 (0%)	1 (17%)
4a	3 (21.5%)	0 (0%)	3 (50%)
N Stage			
0	8 (57%)	8 (100%)	0 (0%)
1	1 (7%)	0 (0%)	1 (17%)
2b	5 (36%)	0 (0%)	5 (83%)
Alcohol consumption			
Drinker	5 (36%)	3 (38%)	2 (33%)
Non-drinker	9 (64%)	5 (62%)	4 (67%)
Smoking or tobacco use			
Ex-smoker	3 (21.5%)	2 (25%)	1 (17%)
Current smoker	3 (21.5%)	1 (12.5%)	2 (33%)
Non-smoker	8 (57%)	5 (62%)	3 (50%)

**Table 2 cells-10-02179-t002:** Top most deregulated candidate EV-associated proteins between healthy controls and oral tongue cancer. Proteins that increase or decrease in expression by 1.5 fold or greater are shaded in green or red respectively. Healthy controls (HC), non-nodal oral tongue squamous cell carcinoma (NN-OTSCC), nodal oral tongue squamous cell carcinoma (N-OTSCC). * Marks fold change that was similar to that observed in head and neck cancer tissues based on TCGA RNA-sequencing data. ^ Marks proteins that were also expressed in head and neck tumor tissues.

Accession	Description	Raw Abundance			Fold Change		
		HC	NN-OTSCC	N-OTSCC	NN-OTSCC Relative to HC	N-OTSCC Relative to HC	N Relative to NN-OTSCC
Potential markers of OTSCC, may also be informative of nodal status							
P10720	Platelet factor 4 variant *	103.9	177.2	54.1	1.7 *	0.5 *	0.3 *
P04350	Tubulin beta-4A chain	66.9	109.7	36.2	1.6	0.5	0.3
Q16778	Histone H2B type 2-E *^,^^	41.4	85.7	21.6	2.1 *	0.5	0.3
P02452	Collagen alpha-1(I) *^,^^	78.8	181.1	41.8	2.3 *	0.5	0.2
Potential markers of OTSCC but not informative on nodal status							
P18827	Syndecan-1 *^,^^	53.8	85.5	78.2	1.6 *	1.5 *	0.9
P08865	40S ribosomal protein SA *^,^^	141.8	77	70.4	0.5 *	0.5	0.9
Potential markers of nodal OTSCC but not non-nodal disease							
P16144-1	Integrin beta-4 ^	83.4	85.2	44.3	1.0	0.5	0.5
P49913	Cathelicidin antimicrobial peptide	110.8	115.3	34.7	1.0	0.3	0.3
Q13201	Multimerin-1 *	116.1	135.2	49.3	1.2	0.4 *	0.4
P02786	Transferrin receptor protein 1 ^	111	93.4	55.7	0.8	0.5	0.6
P04003	C4b-binding protein alpha	128.2	74.4	122.2	0.6	1.0	1.6
P05556-1	Integrin beta-1 ^	76.7	91	49.5	1.2	0.6	0.5
P35579-1	Myosin-9 ^	61.7	88.4	45.2	1.4	0.7	0.5
Q99459	Cell division cycle 5-like Protein ^	124.8	141.4	73.4	1.1	0.6	0.5
O00468	Agrin ^	50.7	69.6	32.5	1.4	0.6	0.5
P26006	Integrin alpha-3 ^	46.1	58.7	31.3	1.3	0.7	0.5
P16070	CD44 antigen ^	68.2	98	45.8	1.4	0.7	0.5
Potential markers of non-nodal OTSCC but not nodal disease							
P35542	Serum amyloid A-4 protein *	87	164.6	72.5	1.9	0.8	0.4 *
P27169	Serum paraoxonase/ arylesterase 1 *^,^^	80.5	178.9	75	2.2	0.9	0.4 *
P09758	Tumour-associated calcium signal transducer 2	61.1	115.1	51.6	1.9	0.8	0.4
P11047	Laminin subunit gamma-1 *^,^^	56.7	90.6	41.1	1.6 *	0.7	0.5
P01116	GTPase Kras ^	51.1	93.7	51.3	1.8 *	1.0	0.5
P12111	Collagen alpha-3(VI) *	74	196.7	54.7	2.7 *	0.7	0.3
P68104	Elongation factor 1 alpha 1 ^	40	137.9	34.1	3.4	0.9	0.2
O15230	Laminin subunit alpha-5 *	62.9	95.3	47.5	1.5 *	0.8	0.5
P11226	Mannose-binding protein C	78.9	138.6	73.8	1.8	0.9	0.5
P06396	Gelsolin ^	73.6	142.3	67.1	1.9	0.9	0.5
P02746	Complement C1q subcomponent subunit B *	58.2	327.9	40.3	5.6 *	0.7	0.1
A1L4H1-1	Soluble scavenger receptor cysteine-rich domain-containing protein SSC5D ^	79.9	154.3	66.1	1.9	0.8	0.4
P55268	Laminin subunit beta-2 ^	53.4	79.3	38.8	1.5	0.7	0.5
P02748	Complement component C9 *	75.8	168.8	75.1	2.2 *	1.0	0.4
P02747	Complement C1q subcomponent subunit C *^,^^	100.6	206.1	75.1	2.0 *	0.7	0.4
Q16777	Histone H2A type 2-C *^,^^	42.9	83	26.3	1.9 *	0.6	0.3
P60709	Actin, cytoplasmic 1 *	45.1	110.3	30.7	2.4 *	0.7	0.3
P0DJI9	Serum amyloid A-2 protein *	74.7	149.3	46.3	2.0	0.6	0.3 *
P20851	C4b-binding protein beta *^,^^	132.8	56.7	127.8	0.4	1.0	2.3 *
P33981	Dual specificity protein kinase TTK *^,^^	180.3	58.1	124.5	0.3	0.7	2.1 *
Q15485-1	Ficolin-2	73.1	120.3	85.6	1.6	1.2	0.7
Q9Y4F5	Centrosomal protein of 170 kDa protein B	67.2	108.8	79.5	1.6	1.2	0.7
P04275	Von Willebrand factor	66	105	69.5	1.6	1.1	0.7
P62937	Peptidyl-prolyl cis-trans isomerase A *^,^^	50.7	80.3	50.8	1.6 *	1.0	0.6
P02751	Fibronectin *	71.6	109.8	86.8	1.5 *	1.2	0.8
P27105	Erythrocyte band 7 integral membrane protein *^,^^	54	78.5	62.5	1.5 *	1.2	0.8

**Table 3 cells-10-02179-t003:** Top most deregulated classical, high abundant, plasma EV-associated proteins between healthy controls and oral tongue cancer. Proteins that increase or decrease in expression by 1.5 fold or greater are shaded in green or red respectively. Healthy controls (HC), non-nodal oral tongue squamous cell carcinoma (NN-OTSCC), nodal oral tongue squamous cell carcinoma (N-OTSCC).

Accession	Description	Raw Abundance			Fold Change		
		HC	NN-OTSCC	N-OTSCC	NN-OTSCC Relative to HC	N-OTSCC Relative to HC	N Relative to NN-OTSCC
De-regulated in all comparison groups							
P02652	Apolipoprotein A-II	75.6	126.3	69.4	1.7	0.9	0.5
P06727	Apolipoprotein A-IV	87.3	185.2	53.6	2.1	0.6	0.3
P01859	Immunoglobulin heavy constant gamma 2	79.5	143.4	46.4	1.8	0.6	0.3
A0A0C4DH29	Immunoglobulin heavy variable 1-3	42.9	157.9	61.1	3.7	1.4	0.4
P01591	Immunoglobulin J chain	106.7	164.2	79.5	1.5	0.7	0.5
P01601	Immunoglobulin kappa variable 1D-16	95.1	181.3	65.1	1.9	0.7	0.4
P06312	immunoglobulin kappa variable 4-1	93.3	143.8	62.3	1.5	0.7	0.4
A0A075B6J9	immunoglobulin lambda variable 2-18	84.1	156.3	50.1	1.9	0.6	0.3
P01709	Immunoglobulin lambda variable 2-8	100	162.5	70.7	1.6	0.7	0.4
A0A075B6I9	Immunoglobulin lambda variable 7-46	78.4	196.3	72.5	2.5	0.9	0.4
P13645	Keratin, type I cytoskeletal 10	72.7	136.5	71.6	1.9	1.0	0.5
P04264	Keratin, type II cytoskeletal 1	60.4	146.2	61.5	2.4	1.0	0.4
P02787	Serotransferrin	77.6	119.4	62.8	1.5	0.8	0.5
P68871	Hemoglobin subunit beta	74.6	117.8	89.1	1.6	1.2	0.8
P01857	Immunoglobulin heavy constant gamma 1	74.9	127.2	80.6	1.7	1.1	0.6
A0A0C4DH31	Immunoglobulin heavy variable 1-18	78.2	130.7	93.9	1.7	1.2	0.7
P02671-1	Fibrinogen alpha chain	63.3	108.8	134.4	1.7	2.1	1.2
P02675	Fibrinogen beta chain	73.1	114.7	136.9	1.6	1.9	1.2
P00738	Haptoglobin	49.6	95.6	133.4	1.9	2.7	1.4
P02679	Fibrinogen gamma chain	70.2	95.3	146.3	1.4	2.1	1.5
O14791	Apolipoprotein L1	91.2	128.8	68.1	1.4	0.7	0.5
P01624	Immunoglobulin kappa variable 3-15	137.4	141.2	67.4	1.0	0.5	0.5

## Data Availability

The data presented in this study are available in Appendix A.

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
