# Peer review of "Proteomic Analysis of Circulating Extracellular Vesicles Identifies Potential Biomarkers for Lymph Node Metastasis in Oral Tongue Squamous Cell Carcinoma"

_cells, 2021, doi:10.3390/cells10092179_

Round 1
Reviewer 1 Report
In this manuscript, Qu and collaborators performed proteomic analysis of extracellular vesicles (EVs) isolated from pooled plasma samples of 3 groups: non-cancer controls, non-nodal oral Tongue Squamous Cell Carcinoma (OTSCC), and nodal OTSCC. Although the study identified 43 proteins differentially expressed among groups, these proteins were identified in pools and there is no validation of the proteins by Western blot either in the pools or in individual samples. Also, there appears to be large amounts of contamination of the EV preps with plasma proteins. The characterization of the EVs prior to proteomics is also insufficient. Moreover, the relevance of each comparison needs to be clearly stated in the result and discussion sections. For example, there is no discussion of the possible significance of proteins that are differentially expressed between non-nodal OTSCC vs non-cancer controls but not between nodal OTSCC vs non-cancer control. Additionally, it would be very helpful if the authors could clearly state what are the list of proteins suggested as biomarkers for lymph node metastasis, as the title suggests, and focus on them in the discussion section. In addition to these general comments, here are specific comments and observations that need to be addressed below:
Comments:
- In methods section 2.2, 100µL of plasma from each patient was used to create a pool sample. However, the amount of pooled plasma (900µL) does not correspond to the number of patients in each group (controls = 10, non-nodal = 8, nodal = 6). Please correct and explain that.
- Double check the OTSS abbreviation in section 2.2 methods, line 106.
- Please include the dilution used for EVs prior to SEC (2.2 methods)
- In the methods section 2.5, please explain the criteria used to load the Western blot. Was it based on amount of protein or number of particles? What was the amount loaded?
- In the 3.1 Results section, there is no information about median age and gender of non-cancer control. Please add them to the text.
- In the section 3.2, please correct the number of particles/mL using superscript for exponential numbers. Ex: 3.09x1011 particles/mL
- Fig 1A gives the impression that all 3 groups were pooled in 1 sample prior to EV isolation. The methods section states that it is not the case. To avoid confusion, please modify the figure including individual arrows from each group pointing towards the SEC column.
- Fig 1B shows protein quantification, but there is no information in the method section about this assay. Please include that information.
- The authors used the measurement of the particles/mL to compare the different groups. However, it is not clear if the amount of sample used prior to EV isolation was the same for all the groups (see comment #1) and the final volume obtained after concentrating the fractions was not stated. Please state the final volume obtained after concentrating the fractions. If it was not the same for all groups, which usually happens when concentrating fractions, the authors should use total amount of protein and total amount of particles to compare the different groups.
- On page 5, line 217-219, the authors state that CD9 may not be identifiable by mass spectrometry analysis, but that is not true. One example is the study of Hoshino et al., 2021 published on Cell Resource showing that CD9 was identified by mass spectrometry analysis in different biological sources, including plasma. Please correct/modify this statement – it may be that it depends on the sensitivity of the mass spectrometry and amount of starting sample.
- Fig 1E with the TEM is not of sufficient quality. The wide view image of the EVs is of overly low resolution and the EVs are strangely black. The zoomed EV is not convincing as an EV.
- The quality of the Western blot for TSG101 is low. Looking at the full blot, it looks like the band shown in the main figure is a non-specific set of bands. Additional EV markers should be done, with convincing banding pattern and darkness – ideally ones that are transmembrane or cytoplasmic. Suggestions for markers are in MISEV2018.
- The degree of contamination with plasma proteins should also be assessed, as suggested by MISEV (e.g. probing for lipoproteins or albumin). It appears that there is a lot of that – even the von Willebrand factor that the authors say is consistent with platelet EVs is an extracellular plasma protein with no transmembrane domain, although it does bind to platelet EVs. Complement and serum amyloid proteins likewise seem like contaminants from plasma protein. Could differences from the pools represent differences in plasma protein contamination?
- Line 287 correct “Intergin signalling”.
- Vesicle is misspelled as “vesical” in 4 different places.
- In section 3.4 the author should summarize the most significant findings of Table 2. It is a long table, and it is very confusing to understand what the comparison is showing. What does it mean if it is elevated in NN-OTSCC vs HC, or N-OTSCC vs HC, and N vs NN? Please explain if you are using the comparison to find biomarkers for cancer, nodal status and what are most relevant proteins.
- In this same section 3.4, it is very confusing to understand why the in-silico analysis was performed if the author clearly stated in the previous sentence, lines 289-290, that “the levels of proteins packaged into plasma derived EVs do not necessarily correlate with their relative intra-cellular concentration or tissue expression”. Please explain why the in-silico analysis is relevant in this case since the overall survival was based on tumor expression and not EV expression.
- The Venn diagrams in Fig 3 are very hard to understand.
- Line 342 correct “asso-ciated”.
- In general, the discussion section needs to focus on the main findings of the paper, that should have been stated in the results section. As it is right now, it is not clear what the main findings are, and the discussion seems unfocused. For example, it is not clear the importance of the von Willebrand factor and Integrin beta-3 for the nodal status of OTSCC patients.
Author Response
Thank you for your thorough and constructive review. We appreciate the time that was taken to review our work and have carefully responded to each point below.
General Comments:
Although the study identified 43 proteins differentially expressed among groups, these proteins were identified in pools and there is no validation of the proteins by Western blot either in the pools or in individual samples.
We agree that our study lacks validation. We have ordered antibodies to candidate proteins and plan to validate in individual samples. However, we have been warned of significant delivery delays (we assume due to Covid-19). Considering the fast pace of the field and the interesting data, we decided to try publish this as a proof of concept, hypothesis-generating study.
Also, there appears to be large amounts of contamination of the EV preps with plasma proteins.
We agree that there is a large amount of plasma protein contamination and have acknowledged this throughout the manuscript. We have attempted to manage this by grouping known plasma proteins and analysing these as a separate group to the candidate proteins of interest.
The characterization of the EVs prior to proteomics is also insufficient.
As requested, we have provided higher resolution TEM images as well. Our NTA data corroborates the TEM results. We have used two positive and one negative marker for WB analyses.
Moreover, the relevance of each comparison needs to be clearly stated in the result and discussion sections. For example, there is no discussion of the possible significance of proteins that are differentially expressed between non-nodal OTSCC vs non-cancer controls but not between nodal OTSCC vs non-cancer control.
We agree and have heavily revised the results and discussion to address this helpful comment.
Additionally, it would be very helpful if the authors could clearly state what are the list of proteins suggested as biomarkers for lymph node metastasis, as the title suggests, and focus on them in the discussion section.
We agree and have made significant changes to the discussion to address this comment.
Specific Comments:
In methods section 2.2, 100µL of plasma from each patient was used to create a pool sample. However, the amount of pooled plasma (900µL) does not correspond to the number of patients in each group (controls = 10, non-nodal = 8, nodal = 6). Please correct and explain that.
We apologise for this error and have corrected section 2.2 to explain it. We made plasma pools using 200µL from each patient, not 100µL as originally indicated. This meant that we had a total volume of 2mL for controls, 1.6mL for non-nodal OTSCC and 1.2mL for nodal OTSCC. From these pools, 900µL was used for SEC.
Double check the OTSS abbreviation in section 2.2 methods, line 106.
Apologies for this typographical error. It has been corrected in the manuscript.
Please include the dilution used for EVs prior to SEC (2.2 methods).
Apologies for this omission. We have added this information to section 2.2. Pooled plasma (900µL) was diluted with 1.1mL of filtered PBS.
In the methods section 2.5, please explain the criteria used to load the Western blot. Was it based on amount of protein or number of particles? What was the amount loaded?
Apologies for this omission. We have amended this section now to explain that we loaded 10µg of proteins.
In the 3.1 Results section, there is no information about median age and gender of non-cancer control. Please add them to the text.
Thank you for highlighting this important oversight. We have added demographical data to section 3.1 to describe the non-cancer controls.
In the section 3.2, please correct the number of particles/mL using superscript for exponential numbers. Ex: 3.09x1011 particles/mL
Apologies for this typographical error. It has been corrected in the manuscript.
Fig 1A gives the impression that all 3 groups were pooled in 1 sample prior to EV isolation. The methods section states that it is not the case. To avoid confusion, please modify the figure including individual arrows from each group pointing towards the SEC column.
Apologies for the confusing element on this figure. It has been amended in the manuscript.
Fig 1B shows protein quantification, but there is no information in the method section about this assay. Please include that information.
Apologies for this important omission. A BCA protein assay was used. We have added this to sections 2.5 and 2.6.
The authors used the measurement of the particles/mL to compare the different groups. However, it is not clear if the amount of sample used prior to EV isolation was the same for all the groups (see comment #1) and the final volume obtained after concentrating the fractions was not stated. Please state the final volume obtained after concentrating the fractions. If it was not the same for all groups, which usually happens when concentrating fractions, the authors should use total amount of protein and total amount of particles to compare the different groups.
We have addressed this comment by revising various sections throughout the manuscript as follows. Although we used the measurement of the particles/mL to compare EV levels between groups, we used total protein for all downstream applications (WB and mass spectrometry). We have revised sections 2.5 and 2.6 to reflect this and have clarified that the BCA protein assay was used. We confirm that the amount of sample that was used prior to EV isolation was the same for all the groups (900µL). The final volume obtained after concentrating the fractions ranged from 100-200µL. We have revised section 2.2 to show this. We have included data regarding the total amount of protein and have now also used this to compare the different groups in section 3.2.
On page 5, line 217-219, the authors state that CD9 may not be identifiable by mass spectrometry analysis, but that is not true. One example is the study of Hoshino et al., 2021 published on Cell Resource showing that CD9 was identified by mass spectrometry analysis in different biological sources, including plasma. Please correct/modify this statement – it may be that it depends on the sensitivity of the mass spectrometry and amount of starting sample.
Thank you for this insightful comment. We have modified this statement accordingly.
Fig 1E with the TEM is not of sufficient quality. The wide view image of the EVs is of overly low resolution and the EVs are strangely black. The zoomed EV is not convincing as an EV.
We apologise for the unacceptable TEM images that were originally provided. We have replaced these with higher quality images.
The quality of the Western blot for TSG101 is low. Looking at the full blot, it looks like the band shown in the main figure is a non-specific set of bands. Additional EV markers should be done, with convincing banding pattern and darkness – ideally ones that are transmembrane or cytoplasmic. Suggestions for markers are in MISEV2018.
We agree that the signal shown by this western blot is low but this is the only result that we are able to present based on the samples that we had available. This particular western blot was loaded with relatively diluted EVs samples because they were raw fractions taken directly from SEC. These EV samples had not been concentrated because we were using this result to confirm that EVs were present in the corresponding fractions.
The degree of contamination with plasma proteins should also be assessed, as suggested by MISEV (e.g. probing for lipoproteins or albumin). It appears that there is a lot of that – even the von Willebrand factor that the authors say is consistent with platelet EVs is an extracellular plasma protein with no transmembrane domain, although it does bind to platelet EVs. Complement and serum amyloid proteins likewise seem like contaminants from plasma protein. Could differences from the pools represent differences in plasma protein contamination?
Thank you for this comment. The degree of contamination was assessed by mass spectrometry. It is possible that some differences between the pools represent differences in plasma protein contamination. We have attempted to manage this by grouping all known plasma proteins separately to the “candidate” EV-associated proteins. In section 3.5 we reported that a lower percentage (37%, 22 out of 59) of these classical plasma proteins changed in abundance between groups when compared with candidate EV proteins. This suggests that plasma proteins were more consistent among the comparison groups.
Line 287 correct “Intergin signalling”.
Apologies for this typographical error. It has been corrected in the manuscript.
Vesicle is misspelled as “vesical” in 4 different places.
Apologies for these typographical errors. They have been corrected throughout the manuscript.
In section 3.4 the author should summarize the most significant findings of Table 2. It is a long table, and it is very confusing to understand what the comparison is showing. What does it mean if it is elevated in NN-OTSCC vs HC, or N-OTSCC vs HC, and N vs NN? Please explain if you are using the comparison to find biomarkers for cancer, nodal status and what are most relevant proteins.
We agree that table 2 is long and have modified the contents to improve interpretation. In addition, we have summarized the most significant findings in section 3.4. We have attempted to emphasise which comparisons have been made to find biomarkers for cancer or nodal status.
In this same section 3.4, it is very confusing to understand why the in-silico analysis was performed if the author clearly stated in the previous sentence, lines 289-290, that “the levels of proteins packaged into plasma derived EVs do not necessarily correlate with their relative intra-cellular concentration or tissue expression”. Please explain why the in-silico analysis is relevant in this case since the overall survival was based on tumor expression and not EV expression.
We are sorry that we did not explain this more clearly. We have modified section 3.4 to address this and explain the results. We performed the in silico analysis in an attempt to show whether there was any similarity between tumour tissue and EV cargo.
The Venn diagrams in Fig 3 are very hard to understand.
We agree and have deleted this figure from the manuscript.
Line 342 correct “asso-ciated”.
Apologies for this typographical error. It has been corrected in the manuscript.
In general, the discussion section needs to focus on the main findings of the paper, that should have been stated in the results section. As it is right now, it is not clear what the main findings are, and the discussion seems unfocused. For example, it is not clear the importance of the von Willebrand factor and Integrin beta-1 for the nodal status of OTSCC patients.
We agree and have heavily revised the discussion to focus on the main findings.
Reviewer 2 Report
Dear Authors,
Your manuscript investigating on blood Extracellular vesicle sub-proteome in patients with oral tongue cancers is pioneering and highly interesting.
However, there are some aspects that I would kindly ask you to clarify.
MAJOR REVISION(s):
- Comparative Statistics: How did you perform the statistical comparison among the three groups? There is no information in Material and Methods. Might you clarify this aspect, please? Similarly, p-values and/or FDR should be reported in tables (and/or in the results section). Do all your proteins present statistical significance?
- Indications about healthy control characteristics (as age, gender, tobacco and drinking history) would be highly appreciated.
MINOR REVISION(s):
- It is not so clear how the 77 EV-associated proteins were selected. Was the criteria based on GO cellular component enrichment analysis or on Vesiclepedia?
Thank you in advance
Regards
Author Response
Thank you for the constructive and helpful review. We were grateful that you found our manuscript pioneering and highly interesting. We respectfully respond to your comments below.
How did you perform the statistical comparison among the three groups? There is no information in Material and Methods. Might you clarify this aspect, please?
We are sorry for not including this important information. Comparison of EV levels between groups was made using the Kruskal–Wallis one-way analysis of variance test. We have added a new section (2.9 Statistical analysis) to describe this.
Similarly, p-values and/or FDR should be reported in tables (and/or in the results section). Do all your proteins present statistical significance?
Apologies for this omission. We have added p-values to results section 3.2 where appropriate. All candidate proteins included in the results section were significant based on a q value of < 0.05. This is described in section 3.3.
Indications about healthy control characteristics (as age, gender, tobacco and drinking history) would be highly appreciated.
Thank you for highlighting this important oversight. We have added demographical data to section 3.1 to describe the healthy controls.
It is not so clear how the 77 EV-associated proteins were selected. Was the criteria based on GO cellular component enrichment analysis or on Vesiclepedia?
The selection of the 77 EV-association proteins is described in section 3.3. We filtered out proteins based on high q-values and incomplete labelling. Classical plasma proteins were analysed as a separate group and this resulted in the shortlist of 77 EV-associated proteins.
Round 2
Reviewer 1 Report
I would have liked to see Western blot validation both of the EV fractions and of the proteomics results, but I think the authors acknowledge appropriately the limitations of their study so I think it's OK.